# Achieving $\tilde{\mathcal{O}}(1/N)$ Optimality Gap in Restless Bandits through Gaussian Approximation

Chen Yan[1]    Weina Wang[2]    Lei Ying[1]

[1]University of Michigan, Ann Arbor    [2]Carnegie Mellon University

{chenyaa, leiying}@umich.edu    weinaw@cs.cmu.edu

## Abstract

We study the finite-horizon Restless Multi-Armed Bandit (RMAB) problem with $N$ homogeneous arms. Prior work has shown that when an RMAB satisfies a non-degeneracy condition, Linear-Programming-based (LP-based) policies derived from the fluid approximation, which captures the mean dynamics of the system, achieve an exponentially small optimality gap. However, it is common for RMABs to be degenerate, in which case LP-based policies can result in a $\Theta(1/\sqrt{N})$ [1] optimality gap per arm. In this paper, we propose a novel Stochastic-Programming-based (SP-based) policy that, under a uniqueness assumption, achieves an $\tilde{\mathcal{O}}(1/N)$ optimality gap for degenerate RMABs. Our approach is based on the construction of a Gaussian stochastic system that captures not only the mean but also the variance of the RMAB dynamics, resulting in a more accurate approximation than the fluid approximation. We then solve a stochastic program for this system to obtain our policy. This is the first result to establish an $\tilde{\mathcal{O}}(1/N)$ optimality gap for degenerate RMABs.

## 1    Introduction

The Restless Multi-Armed Bandit (RMAB) problem is an important framework in sequential decision-making, where a decision maker selects a subset of tasks (arms) to work on (pull) at each time step to maximize cumulative rewards, under known model parameters [29]. Unlike the classical (restful) bandit [11], in the restless variant, the state of each arm evolves stochastically regardless of whether it is pulled. RMABs have been widely applied in domains such as machine maintenance [9, 12], healthcare resource allocation [22, 23], and target tracking [19, 21], to name a few, where optimal decision-making under uncertainty is critical. A general RMAB is PSPACE-hard [25], and finding optimal policies is computationally challenging, especially as the number of arms grows. Recently, there have also been efforts to use deep learning and reinforcement learning to learn heuristic policies for RMABs, such as [1, 18, 24, 30, 31].

In this paper, we focus on the finite-horizon version of the RMAB problem with $N$ homogeneous arms and horizon $H$, where each arm follows the same (time-dependent, known) state transition and reward function. While computing the exact optimal policy is impractical, the homogeneity of the model allows for the design of computationally efficient policies. One such class of policies is

---

[1]We adopt standard asymptotic notation throughout this paper. Specifically, for functions $f(N)$ and $g(N)$, we write $f(N) = \mathcal{O}(g(N))$ if there exist positive constants $C$ and $N_0$ such that $|f(N)| \le C|g(N)|$ for all $N \ge N_0$. Similarly, we write $f(N) = \Omega(g(N))$ if $g(N) = \mathcal{O}(f(N))$, and $f(N) = \Theta(g(N))$ if both $f(N) = \mathcal{O}(g(N))$ and $f(N) = \Omega(g(N))$ hold simultaneously. Additionally, we use $\tilde{\mathcal{O}}(\cdot)$ and $\tilde{\Theta}(\cdot)$ notation to indicate that logarithmic factors are omitted.

39th Conference on Neural Information Processing Systems (NeurIPS 2025).

based on fluid approximation, which transforms the original $N$-armed RMAB problem into a Linear Program (LP), and an LP-based policy can be efficiently computed based on the solution to the LP.

**Optimality gap.** In [15], an LP-based index policy was proposed and achieves an $o(1)$ optimality gap. This gap was later improved to $\mathcal{O}(\log N/\sqrt{N})$ in [34], and subsequently to $\mathcal{O}(1/\sqrt{N})$ in [7]. In the numerical experiments of [7], it was observed that while this $\mathcal{O}(1/\sqrt{N})$ gap appears tight for certain problems, in others the gap converges to zero more rapidly. This empirical observation was theoretically confirmed in [36], where it was shown that under a *non-degenerate* condition (formally defined in Definition 3.1), the gap is at a smaller order of $\mathcal{O}(1/N)$.

**Non-degenerate condition.** This non-degenerate condition has since become a key assumption in subsequent works. [10] shows that, under this condition, the optimality gap becomes exponentially small when the rounding error induced by scaling the fluid approximation by $N$ is eliminated. Further generalizations of the non-degenerate condition have been made in [8], extending it to multi-action and multi-constraint RMABs (also known as weakly coupled Markov decision processes). In [35], the non-degenerate condition was further extended to settings with heterogeneous arms.

**Prevalence of degenerant RMABs.** Despite the central role played by the non-degenerate condition, many RMAB problems are *degenerate*. Notable examples, which originate from real-world applications, have been discussed in [7, 8, 36]. Moreover, a numerical study presented in Appendix B.1 revealed that a significant proportion (about 50% for some cases) of randomly generated RMABs are degenerate, and almost all of them satisfy the Uniqueness Assumption 4.1. This highlights the practical importance of addressing degenerate RMABs.

However, to the best of our knowledge, in all previous works, when an RMAB is *degenerate*, the best known optimality gap is $\mathcal{O}(1/\sqrt{N})$. It is widely believed that this upper bound is also order-wise tight under LP-based policies, making it $\Theta(1/\sqrt{N})$. We will formally prove this result in Theorem 4.2 by using an example.

These results led us to ask the following central question of this paper:

*Does there exist a computationally efficient algorithm for degenerate RMABs with an optimality gap order-wise smaller than $\mathcal{O}(1/\sqrt{N})$?*

**Contributions.** This paper answers the question affirmatively and includes the following results:

- We construct a Gaussian stochastic system (see (14)) that more accurately captures the behavior of the $N$-system than the fluid system. Unlike the fluid system, which serves as a first-order approximation to the stochastic $N$-system, the Gaussian stochastic system incorporates both the mean and variance of the $N$-system.

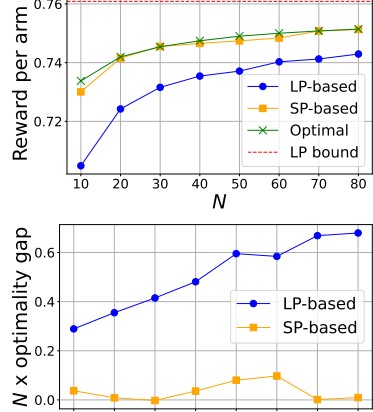

- We demonstrate that the SP-based policy obtained from the Gaussian stochastic system (see Algorithm (1)) achieves $\tilde{\mathcal{O}}(1/N)$ optimality gap for degenerate RMABs that satisfy the Uniqueness Assumption 4.1 (see Theorem 4.1). We further prove that for degenerate RMABs without the Uniqueness Assumption, SP-based policy results in $\Omega(1/\sqrt{N})$ improvement per arm compared with a large class of LP-based policies (see Theorem 4.3).

Figure 1: Comparison of LP and SP-based policies.

- We further compliment our main result by presenting a *degenerate* example in Section 3, demonstrating in Theorem 4.2 that not only the LP-based policy has $\Theta(1/\sqrt{N})$ optimality gap, but also the LP upper bound has $\Theta(1/\sqrt{N})$ gap from the optimal value. This indicates that the LP upper bound, which is a widely used baseline, itself is not tight for degenerate RMABs.

As an illustration, consider a degenerate RMAB example with 2 states and a horizon of 2 steps (the details can be found in Section 3.3). Given the small problem size, the optimal policy can be computed exactly through brute-force methods. As shown in Figure 1, the performance of the policy obtained by solving the Gaussian stochastic system is already very close to the true optimal value.

Considering $N \times$ optimality gap (i.e. the total optimality gap of the $N$ arms with respect to the optimal policy), it remains bounded under the SP-based policy as $N$ increases, which confirms the $\tilde{\mathcal{O}}(1/N)$ optimality gap; while the total optimality gap for the LP-based policy appears to grow unbounded.

**Related work.** A comparison of state-of-the-art results in finite-horizon RMABs with this work is provided in Table 1.

Table 1: Optimality gap and assumptions in finite-horizon RMABs

| Paper | Gap | Assumption |
|---|---|---|
| Brown and Smith [7] | $\mathcal{O}(1/\sqrt{N})$ | General |
| Gast et al. [10] | $\mathcal{O}(\exp(-CN))$ | Non-degeneracy |
| This Work | $\tilde{\mathcal{O}}(1/N)$ | Degeneracy & Uniqueness Assumption 4.1 |

Gaussian approximations, rooted in the central limit theorem, are widely used to approximate stochastic processes. In heavy-traffic queueing analysis, this yields the *diffusion approximation*, where the centered, scaled limit is a diffusion process; see early work in [4, 5, 16, 17]. This literature typically studies *fixed policies* and establishes convergence (or rates) to the diffusion limit. In contrast, we aim to *design* near-optimal policies for discrete-time RMABs via a second-order Gaussian approximation.

Closer to our setting is approximate diffusion control (e.g., [2, 13, 14]), which replaces the original problem with a diffusion control problem solved through the Hamilton–Jacobi–Bellman (HJB) equation. Recent work [6] shows HJBs arise from second-order value-function approximations of general MDPs. However, solving HJBs is notoriously difficult, especially when diffusion coefficients are control dependent, yielding highly nonlinear PDEs. This actually motivates our alternative second-order approach, detailed in Section 3.

**Notational convention.** Vectors are row vectors by default. Throughout the paper, we consider three systems: the $N$-armed RMAB system, the fluid system, and the Gaussian stochastic system. As a general convention, variables and functions with " ¯ " are for the fluid system, while those with " ˜ " are for the Gaussian stochastic system. The terms "action" and "control" are used interchangeably in the context of decision-making.

## 2 Problem formulation

**RMAB model.** We consider the $H$-horizon Restless Multi-Armed Bandit (RMAB) problem with $N$ homogeneous arms. Each arm is modeled as a Markov Decision Process (MDP) with state space $\mathcal{S} := \{1, 2, \ldots, S\}$ and action space $\mathcal{A} := \{0, 1\}$. At each time step $h$ ($1 \leq h \leq H$), the decision maker decides which arms to take action 1, also referred to as the pulling action, subject to the budget constraint that exactly $\alpha N$ arms should be pulled. Here $0 < \alpha < 1$ and we assume $\alpha N$ is an integer. After the actions are applied, the $N$ arms evolve independently. Specifically, the state transitions from $\mathbf{s} \in \mathcal{S}^N$ to $\mathbf{s}' \in \mathcal{S}^N$ with probability $\mathbf{P}_h(\mathbf{s}' \mid \mathbf{s}, \mathbf{a}) = \prod_{n=1}^{N} \mathbf{P}_h(s'_n \mid s_n, a_n)$, where $(s_n, a_n)$ denotes the state-action pair for the $n$-th arm, and $\mathbf{P}_h(\cdot \mid \cdot, a)$ is the transition kernel under action $a$. For convenience, we refer to this RMAB system with $N$ arms as the $N$-*system*.

At each time step $h$, the decision maker collects an additive reward $\sum_{n=1}^{N} r_h(s_n, a_n)$, where $r_h(s, a)$ denotes the reward for the state-action pair $(s, a)$ at time $h$ and is assumed to be nonnegative without loss of generality. The objective is to find a policy $\pi^N$, which maps the bandit state vector $\mathbf{s}_h = (s_{1,h}, s_{2,h}, \ldots, s_{N,h})$ to an action vector $\mathbf{a}_h = (a_{1,h}, a_{2,h}, \ldots, a_{N,h})$ for each $h$, that maximizes the total expected reward per arm over the horizon:

$$V_{\text{opt}}^N := \max_{\pi^N} \quad \sum_{h=1}^{H} \frac{1}{N} \sum_{n=1}^{N} \mathbb{E}_{\pi^N} \left[ r_h(s_{n,h}, a_{n,h}) \right] \tag{1}$$

$$\text{s.t.} \quad \sum_{n=1}^{N} a_{n,h} = \alpha N, \quad 1 \leq h \leq H. \tag{2}$$

To reduce the computational complexity, we leverage the fact that the $N$ arms are homogeneous and aggregate the states of individual arms. This gives us a simplified representation of the bandit state, which facilitates subsequent analysis and policy design. Specifically, we represent the bandit state at time $h$ as a vector $\mathbf{X}_h = (X_h(s))_{s\in\mathcal{S}} \in \Delta_N^S$, where each $X_h(s)$ denotes the fraction of arms in state $s \in \mathcal{S}$. Here, $\Delta_N^S$ is a discrete subset of $\Delta^S$, the probability simplex over $\mathcal{S}$, such that each $\mathbf{x} \in \Delta_N^S$ satisfies that $N\mathbf{x}$ has all integer entries. Similarly, we represent the action at time $h$ as a vector $\mathbf{Y}_h = (Y_h(s,a))_{s\in\mathcal{S},a\in\mathcal{A}} \in \Delta_N^{2S}$, where each $Y_h(s,a)$ denotes the fraction of arms in state $s$ taking action $a$. We treat $\mathbf{Y}_h$ as a row vector of length $2S$. We also write $\mathbf{Y}_h(\cdot,0) = (Y_h(s,0))_{s\in\mathcal{S}}$ and $\mathbf{Y}_h(\cdot,1) = (Y_h(s,0))_{s\in\mathcal{S}}$, which are both row vectors of length $S$.

Under the new state and action representation, given an action $\mathbf{Y}_h = \mathbf{y}_h$, the state evolves as

$$\mathbf{X}_{h+1} = \frac{1}{N}\sum_{s,a}\mathbf{U}_h^{(s,a)}(\mathbf{y}_h), \tag{3}$$

where $\mathbf{U}_h^{(s,a)}(\mathbf{y}_h) \in \mathbb{N}^S$ follows a multinomial distribution $\mathrm{multi}(Ny_h(s,a), \mathbf{P}_h(\cdot \mid s,a))$, and the $\mathbf{U}_h^{(s,a)}(\mathbf{y}_h)$'s for different $(s,a)$'s are *independent*.

Under the new state and action representation, a policy $\pi^N$ maps the state $\mathbf{X}_h$ to an action vector $\mathbf{Y}_h$ for each $h$. We also rewrite the reward function as a vector $\mathbf{r}_h = (r_h(s,a))_{s\in\mathcal{S},a\in\mathcal{A}} \in \mathbb{R}^{2S}$. Suppose the initial state of the $N$-system is $\mathbf{X}_1 = \mathbf{x}_{\mathrm{ini}} \in \Delta_N^S$. Then the objective of the RMAB problem can be reformulated as:

$$V_{\mathrm{opt}}^N(\mathbf{x}_{\mathrm{ini}}, 1) = \max_{\pi^N} \sum_{h=1}^H \mathbb{E}_{\pi^N}\left[\mathbf{r}_h\mathbf{Y}_h^\top\right] \tag{4}$$

$$\text{s.t.} \quad \sum_s Y_h(s,1) = \alpha, \quad 1 \le h \le H, \tag{5}$$

$$\mathbf{Y}_h(\cdot,0) + \mathbf{Y}_h(\cdot,1) = \mathbf{X}_h, \quad \mathbf{Y}_h \in \Delta_N^{2S}, \quad 1 \le h \le H. \tag{6}$$

Here (5) corresponds to the budget constraint, and (6) ensures that $\mathbf{Y}_h$ is a valid action for state $\mathbf{X}_h$.

**Fluid approximation.** One classical approach to address the complexity of RMAB is to consider a fluid approximation as in [7, 10, 15, 34, 36], where the stochastic state transition is replaced by its expected value, leading to a deterministic transition at each step:

$$\mathbf{x}_{h+1} = \sum_{s,a} y_h(s,a)\mathbf{P}_h(\cdot \mid s,a). \tag{7}$$

This fluid system smooths out the stochastic fluctuations in the $N$-system and leads to the following Linear Program (LP), which we refer to as the *fluid LP*:

$$\overline{V}_{\mathrm{LP}}(\mathbf{x}_{\mathrm{ini}}, 1) := \max_{(\mathbf{x}_h,\mathbf{y}_h)_{h\in\{1,2,\ldots,H\}}} \sum_{h=1}^H \mathbf{r}_h\mathbf{y}_h^\top \tag{8}$$

$$\text{s.t.} \quad \sum_s y_h(s,1) = \alpha, \quad 1 \le h \le H, \tag{9}$$

$$\mathbf{y}_h(\cdot,0) + \mathbf{y}_h(\cdot,1) = \mathbf{x}_h(\cdot), \quad \mathbf{y}_h \ge \mathbf{0}, \quad 1 \le h \le H, \tag{10}$$

$$\mathbf{x}_1 = \mathbf{x}_{\mathrm{ini}}, \quad \mathbf{x}_{h+1}(\cdot) = \sum_{s,a} y_h(s,a)\mathbf{P}_h\left(\cdot \mid s,a\right), \quad 1 \le h \le H-1. \tag{11}$$

Let $\mathbf{x}^* = (\mathbf{x}_h^*)_{h\in\{1,2,\ldots,H\}}$ and $\mathbf{y}^* = (\mathbf{y}_h^*)_{h\in\{1,2,\ldots,H\}}$ be an optimal solution to this fluid LP. Note that this LP can be efficiently solved. Furthermore, it has been shown that $V_{\mathrm{opt}}^N(\mathbf{x}_{\mathrm{ini}}, 1) \le \overline{V}_{\mathrm{LP}}(\mathbf{x}_{\mathrm{ini}}, 1)$; see, for instance, [36, Lemma 1] or [32, Lemma 3]. That is, the optimal value of the LP provides an upper bound on the optimal value of the RMAB problem. Based on the solution to this LP, LP-based policies can be obtained but have $\Theta(1/\sqrt{N})$ optimality gap per arm in degenerate RMABs, as we pointed out in the Introduction.

# 3 Gaussian approximation and SP-based policy

## 3.1 Gaussian stochastic system

The performance of LP-based policies is inherently limited by how accurately the fluid system approximates the original $N$-system. To design policies that outperform the LP-based policies, we construct a Gaussian stochastic system that better approximates the original $N$-system by capturing not only the mean but also the variance of the system. This Gaussian stochastic system is centered around $\mathbf{y}^*$, an optimal solution to the fluid LP. We then search for an optimal policy for the Gaussian stochastic system in a neighborhood of $\mathbf{y}^*$, which adjusts $\mathbf{y}^*$ to account for the stochasticity. This policy is then applied to the $N$-system after properly handling the integer effect.

Specifically, we construct a Gaussian stochastic system with state space $\Delta^S$ and action space $\Delta^{2S}$. The system has the same initial state $\mathbf{x}_{\mathrm{ini}}$ as the $N$-system. Under action vector $\mathbf{y}_h$, the state of the Gaussian stochastic system at the next time step, $\tilde{\mathbf{X}}_{h+1}$, is a random vector of the following form:

$$\tilde{\mathbf{X}}_{h+1} = \mathrm{Proj}_{\Delta^S}\left(\sum_{s,a} y_h(s,a)\mathbf{P}_h(\cdot \mid s,a) + \mathbf{Z}_h/\sqrt{N}\right), \tag{12}$$

where $\mathbf{Z}_h$ is a Gaussian random vector with distribution $\mathcal{N}(\mathbf{0}, \Gamma_h(\mathbf{y}_h^*))$, and $\mathbf{Z}_h$'s are independent across steps. The covariance matrix $\Gamma_h(\mathbf{y}_h^*)$ is a constant matrix independent of $N$, with its explicit expression given in Appendix A. The projection, $\mathrm{Proj}_{\Delta^S}(\cdot)$, is onto the simplex $\Delta^S$ under the $\ell_2$-distance. This projection will not be used with a high probability. Indeed, it can be shown that $\sum_{s,a} y_h(s,a)\mathbf{P}_h(\cdot \mid s,a) + \mathbf{Z}_h/\sqrt{N} \in \Delta^S$ occurs with probability $1 - \mathcal{O}(1/N^{\log N})$, see Lemma C.8 of [33] for a proof.

We now explain how the Gaussian stochastic system captures the mean and variance of the $N$-system. Suppose the action at time $h$ is $\mathbf{y}_h$. Ignoring the projection, we have $\tilde{\mathbf{X}}_{h+1} = \sum_{s,a} y_h(s,a)\mathbf{P}_h(\cdot \mid s,a) + \mathbf{Z}_h/\sqrt{N}$. The term $\sum_{s,a} y_h(s,a)\mathbf{P}_h(\cdot \mid s,a)$ is the expectation of the state $\mathbf{X}_{h+1}$ in the $N$-system, which is the same as the state $\mathbf{x}_{h+1}$ in the fluid system under action $\mathbf{y}_h$. The additional term, $\mathbf{Z}_h/\sqrt{N}$, is random, and by construction, its covariance matrix matches that of $\mathbf{X}_{h+1}$ in the $N$-system if $\mathbf{y}_h = \mathbf{y}_h^*$. Therefore, this term captures the variance of the $N$-system when $\mathbf{y}_h$ is close to the optimal solution $\mathbf{y}_h^*$ of the fluid LP.

We remark that a key innovation of our approach is to replace the otherwise (state,action)-dependent diffusion terms with (state,action)-*independent* ones. We construct the Gaussian system so that the covariance of the noise is fixed—chosen from the fluid-optimal solution $\mathbf{y}_h^*$—rather than depending on $\mathbf{y}_h$. This enables the use of stochastic-programming techniques that require action-independent randomness, such as the EDDP algorithm [20, Algorithm 3] we employ in Section 5. Conceptually, this is a further "simplification" of the classical diffusion approximation: although the covariance is fixed at $\mathbf{y}_h^*$, the resulting approximation retains the *same order* of error for our problem (a point established in Theorem 4.1) while remaining computationally tractable.

This simplification is justified by scale separation: in the $N$-system, stochastic fluctuations are $\mathcal{O}(1/\sqrt{N})$ relative to the deterministic drift (see the $1/\sqrt{N}$ factor multiplying $\mathbf{Z}_h$ in (12)). Hence the covariance induced by the fluid-optimal action serves as a robust *surrogate* for the action-dependent covariance without changing the asymptotic accuracy, a property that typically does not hold in traditional diffusion control but is available here due to the central limit theorem scaling.

## 3.2 Stochastic-Programming-based (SP-based) policy

After constructing the Gaussian stochastic system, we search for an optimal policy in the Gaussian stochastic system. However, we restrict the search to a *neighborhood of* $\mathbf{y}^*$, since the Gaussian stochastic system closely approximates the $N$-system when the state and action of the $N$-system stay close to $\mathbf{y}^*$. In particular, given a fixed parameter $\delta_N := 2\log N/\sqrt{N} = \tilde{\Theta}(1/\sqrt{N})$, we define the following policy class:

$$\begin{aligned}\Pi_{\delta_N}(\mathbf{y}^*) := \big\{\pi \colon \forall 1 \le h \le H, \ \|\pi(\mathbf{x}_h, h) - \mathbf{y}_h^*\|_\infty \le \kappa z_h \delta_N, \ \text{if } \|\mathbf{x}_h - \mathbf{x}_h^*\|_\infty \le z_h \delta_N; \\ \pi = \pi_{\mathrm{pre}} \text{ for a predefined policy } \pi_{\mathrm{pre}}, \ \text{if } \|\mathbf{x}_h - \mathbf{x}_h^*\|_\infty > z_h \delta_N\big\}.\end{aligned} \tag{13}$$

---

**Algorithm 1** Stochastic-Programming-based (SP-based) policy

---

1: **Input:** An optimal solution $\mathbf{y}^*$ to LP (8); constants $z_h$, $\delta_N$ and $\kappa$; a predefined policy $\pi_{\text{pre}}$
2: **if** $\mathbf{y}^*$ is non-degenerate **then**
3:     Use an LP-based policy
4:     **Break**
5: **end if**
6: Solve the Gaussian stochastic program (14) to obtain an optimal policy $\tilde{\pi}^{N,*} \in \Pi_{\delta_N}(\mathbf{y}^*)$
7: **for** $h = 1$ **to** $H$ **do**
8:     $\mathbf{x}_h \leftarrow$ state of the $N$-system at time $h$
9:     **if** $\|\mathbf{x}_h - \mathbf{x}_h^*\|_\infty \leq z_h \delta_N$ **then**
10:         $\mathbf{Y}_h \leftarrow \text{round}(\tilde{\pi}^{N,*}(\mathbf{x}_h, h))$
11:     **else**
12:         $\mathbf{Y}_h \leftarrow \text{round}(\pi_{\text{pre}}(\mathbf{x}_h, h))$
13:     **end if**
14:     Apply action $\mathbf{Y}_h$
15: **end for**

---

Here $\kappa$ is a positive constant with value $\kappa := \max\{2 + 6S, 3 + 2r_{\max}HS/\sigma\}$, where $r_{\max} := \max_{s,a,h} r_h(s,a)$; $\sigma$ is a constant depending on the fluid LP; $z_h$ for $1 \leq h \leq H$ is a recursive sequence. The explicit expressions for these constants are collected in Appendix A of [33]. We note that all of them are independent of $N$, and can be computed or estimated solely from the fluid LP (8).

We now explain the reasoning behind the definition of the policy class $\Pi_{\delta_N}(\mathbf{y}^*)$. We restrict corrections to a $\widetilde{\mathcal{O}}(1/\sqrt{N})$ neighborhood of the fluid-optimal state–action. When the LP has a unique solution, an optimal $N$-system policy lies within this neighborhood (see Lemma C.1 of [33] for a proof); hence our restriction retains optimal policies while ensuring that the second-order approximation incurs only $\widetilde{\mathcal{O}}(1/N)$ error (see Lemma C.4 of [33] for a proof). Enlarging the neighborhood inflates the approximation error, whereas shrinking it risks excluding the true optimum—so $\widetilde{\mathcal{O}}(1/\sqrt{N})$ is the "right" scale for our method. Moreover, since the smallest gap of interest is $\widetilde{\mathcal{O}}(1/N)$ and the process leaves this neighborhood with probability $\mathcal{O}(1/N^{\log N})$ (Lemma C.8 of [33]), the contribution of out-of-neighborhood behavior is negligible. Consequently, we do not distinguish between policies optimized only locally and those also optimized outside the neighborhood; whenever the state exits, we simply follow a predefined policy as in (13).

We then formulate the following Gaussian stochastic program (SP):

$$\max_{\pi \in \Pi_{\delta_N}(\mathbf{y}^*)} \quad \sum_{h=1}^{H} \mathbb{E}\left[\mathbf{r}_h \tilde{\mathbf{Y}}_h^\top\right] \tag{14}$$

$$\text{s.t.} \quad \sum_s \tilde{Y}_h(s,1) = \alpha, \quad 1 \leq h \leq H, \tag{15}$$

$$\tilde{\mathbf{Y}}_h(\cdot,0) + \tilde{\mathbf{Y}}_h(\cdot,1) = \tilde{\mathbf{X}}_h, \quad \tilde{\mathbf{Y}}_h \geq \mathbf{0}, \quad 1 \leq h \leq H, \tag{16}$$

$$\tilde{\mathbf{X}}_{h+1} = \text{Proj}_{\Delta^S}\left(\sum_{s,a} \tilde{Y}_h(s,a)\mathbf{P}_h(\cdot \mid s,a) + \mathbf{Z}_h/\sqrt{N}\right), \quad 1 \leq h \leq H-1. \tag{17}$$

We now present our SP-based policy, formally described in Algorithm 1. The core idea of this policy is to solve the Gaussian stochastic program and obtain an optimal policy $\tilde{\pi}^{N,*} \in \Pi_{\delta_N}(\mathbf{y}^*)$. This policy is then applied to the $N$-system through a simple rounding procedure. It guarantees that the action generated by the algorithm pulls an *integer* number of agents. Since when computing the first- and second-order approximations, we extend the state and action spaces from their original discrete sets (with granularity $1/N$) to continuous-valued probability simplices. The action computed in this continuous domain must subsequently be converted back to the discrete domain, ensuring that all values multiplied by $N$ are integers and thus applicable to the $N$-agent problem. This rounding procedure is explicitly detailed in Appendix A, in which we showed alongside that the rounding error is of order $\mathcal{O}(1/N)$.

We remark that we only use the policy $\tilde{\pi}^{N,*}$ when the RMAB is degenerate. When the RMAB is non-degenerate, our policy defaults to a LP-based policy, which prior work (see Introduction) has shown to achieve an exponentially small optimality gap (in terms of $N$) relative to $\overline{V}_{\mathrm{LP}}$. The definition of non-degeneracy is given below, and it is easy for an algorithm to check whether an RMAB is non-degenerate or not.

**Definition 3.1** (Non-degeneracy [8, 10, 36]). *An RMAB is* non-degenerate *if, its corresponding fluid LP (8) admits an optimal solution $\mathbf{y}^*$, such that for each $h$ with $1 \le h \le H$, there exists at least one state $s \in \mathcal{S}$ such that $y_h^*(s,0) > 0$ and $y_h^*(s,1) > 0$.*

### 3.3 Illustration of the SP-based policy on a degenerate example

We illustrate the SP-based policy via a two-state RMAB with horizon $H = 2$ and pulling budget $\alpha = 0.5$. The rewards are given by $r_1(1,1) = r_2(1,1) = 1$, with all other $(h,s,a)$-tuples being $r_h(s,a) = 0$. The transition probabilities at $h = 1$ are $\mathbf{P}_1(1 \mid 1,1) = 0.2$, $\mathbf{P}_1(1 \mid 1,0) = 0.9$, $\mathbf{P}_1(1 \mid 2,1) = 0.7$, $\mathbf{P}_1(1 \mid 2,0) = 0.25$. Assume at $h = 1$ there are $N/2$ arms in state 1, and the other $N/2$ arms are in state 2.

**$N$-system problem.** Note that at time $h = 2$, only $r_2(1,1) = 1$, so the optimal action at $h = 2$ is to pull as many arms in state 1 as possible, i.e., $Y_2(1,1) = \min\{0.5, X_2(1)\}$. Therefore, we only need to decide the optimal action at $h = 1$. We further notice that since $X_1(1) = X_1(2) = \alpha = 0.5$, the entries of $\mathbf{Y}_1$ are all determined by $Y_1(1,1)$ as follows

$$Y_1(1,0) = Y_1(2,1) = 0.5 - Y_1(1,1), \quad Y_1(2,0) = Y_1(1,1).$$

Therefore, we only need to optimize $Y_1(1,1)$. The $N$-system optimal policy is given by the solution of the following problem

$$\max_{0 \le Y_1(1,1) \le 0.5} Y_1(1,1) + \mathbb{E}\left[\min\{0.5, X_2(1)\}\right]. \tag{18}$$

**Fluid LP and its optimal solution.** In the fluid system, we replace $X_2(1)$ with its mean $x_2(1) = \sum_{s,a} y_1(s,a)\mathbf{P}_1(1 \mid s,a) = 0.8 - 1.15 \times y_1(1,1)$. Then the fluid LP is

$$\overline{V}_{\mathrm{LP}} = \max_{0 \le y_1(1,1) \le 0.5} y_1(1,1) + \min\{0.5, \ 0.8 - 1.15 \times y_1(1,1)\}, \tag{19}$$

which exchanges the expectation and the min operator in the $N$-system problem (18). The optimal solution is $y_1^*(1,1) = 0.2609$. One can verify that this problem is degenerate (see Definition 3.1).

**SP-based policy.** Given the fluid solution $\mathbf{y}^*$ above, the corresponding Gaussian stochastic program is

$$\max_{0 \le \tilde{Y}_1(1,1) \le 0.5} \tilde{Y}_1(1,1) + \mathbb{E}\left[\min\left\{0.5, \ 0.8 - 1.15 \times \tilde{Y}_1(1,1) + \frac{Z_1}{\sqrt{N}}\right\}\right], \tag{20}$$

where $Z_1 \overset{d}{\sim} \mathcal{N}(0, 0.1624)$. Since we are searching for an optimal solution around $\mathbf{y}_1^*$, let $\tilde{Y}_1(1,1) = y_1^*(1,1) + \frac{c}{\sqrt{N}}$. Then the stochastic program can be written as

$$\mathbf{y}_1^*(1,1) + 0.5 + \frac{1}{\sqrt{N}} \max_c \left(c + \mathbb{E}\left[\min\{0, Z_1 - 1.15c\}\right]\right),$$

which is equivalent to the following problem

$$\max_c \ c + \mathbb{E}\left[\min\{0, Z_1 - 1.15c\}\right].$$

There exists an explicit and unique solution to the problem above, and the solution can be numerically computed and is $c_{\mathrm{d}}^* = 0.3940$ (for more details please refer to Appendix D of [33]). Therefore, the SP-based policy is given by $\mathrm{round}(\tilde{Y}_1^*(1,1)) = \mathrm{round}(0.2609 + 0.3940/\sqrt{N})$. This SP-based policy outperforms LP-based policies, as illustrated in Figure 1 in the Introduction.

**Some insights.** We compare the fluid approximation in (19) with the Gaussian approximation in (20) when they both take an action $y_1(1,1) = \tilde{Y}_1(1,1) = y_1^*(1,1) + c/\sqrt{N}$ for a positive constant $c$. In the fluid system, the reward for $h = 2$ is $\min\{0.5,\ 0.8 - 1.15 \times y_1(1,1)\}$, which is capped at $0.5$. One can verify that this deviates from the value $\mathbb{E}\left[\min\{0.5, X_2(1)\}\right]$ in the $N$-system by $\Theta(1/\sqrt{N})$, caused by the exchange of the expectation and the min operator. In contrast, in the Gaussian stochastic system,

$$\mathbb{E}\left[\min\left\{0.5, 0.8 - 1.15 \times \tilde{Y}_1(1,1) + \frac{Z_1}{\sqrt{N}}\right\}\right] = 0.5 + \mathbb{E}\left[\min\left\{0, \frac{Z_1 - 1.15c}{\sqrt{N}}\right\}\right],$$

which can be verified to be $\tilde{\mathcal{O}}(1/N)$ away from the value $\mathbb{E}\left[\min\{0.5, X_2(1)\}\right]$ in the $N$-system and thus giving a better approximation. This better approximation allows us to find a better policy near $y_1^*(1,1)$. The inaccuracy of the fluid approximation is in fact a fundamental reason that LP-based policies have a $\Theta(1/\sqrt{N})$ optimality gap for some degenerate RMABs, whereas the correction in our SP-based policy reduces the $\Theta(1/\sqrt{N})$ inaccuracy to $\tilde{\mathcal{O}}(1/N)$.

# 4 Main theoretical results

In this section, we present our main theoretical results. We will frequently use the following quantities. Let $V_\pi^N(\mathbf{x}_h, h)$ and $Q_\pi^N(\mathbf{x}_h, \mathbf{y}_h, h)$ denote the value function and Q-function of policy $\pi$ evaluated in the $N$-system; and $\tilde{V}_\pi^N(\mathbf{x}_h, h)$ and $\tilde{Q}_\pi^N(\mathbf{x}_h, \mathbf{y}_h, h)$ denote the value function and Q-function of a policy $\pi$ evaluated in the Gaussian stochastic system. Further, we consider the following optimal policies within the policy class $\Pi_{\delta_N}(\mathbf{y}^*)$:

$$\tilde{\pi}^{N,*} \in \underset{\pi \in \Pi_{\delta_N}(\mathbf{y}^*)}{\arg\max}\ \tilde{V}_\pi^N(\mathbf{x}_h, h),$$

which is referred to as the *locally-SP-optimal* policy. We sometimes say that we apply the locally-SP-optimal policy $\tilde{\pi}^{N,*}$ to the $N$-system and denote its value function as $V_{\tilde{\pi}^{N,*}}^N(\mathbf{x}_h, h)$, with the understanding that we apply $\tilde{\pi}^{N,*}$ with the rounding procedure, detailed in Appendix A.

## 4.1 Global optimality

**Assumption 4.1** (Uniqueness). *The fluid LP in* (8) *has a unique optimal solution* $\mathbf{y}^*$.

Note that once we obtain an optimal solution to the LP, verifying uniqueness is straightforward [3].

**Theorem 4.1** (Global optimality). *Consider an RMAB that satisfies the Uniqueness Assumption 4.1. Then the locally-SP-optimal policy, $\tilde{\pi}^{N,*}$, when applied to the $N$-system (with rounding), achieves an optimality gap of $\tilde{\mathcal{O}}(1/N)$; i.e.,*

$$V_{\mathrm{opt}}^N(\mathbf{x}_{\mathrm{ini}}, 1) - V_{\tilde{\pi}^{N,*}}^N(\mathbf{x}_{\mathrm{ini}}, 1) = \tilde{\mathcal{O}}(1/N),$$

*where $V_{\mathrm{opt}}^N$ is the optimal value function, and $V_{\tilde{\pi}^{N,*}}^N$ is the value function of $\tilde{\pi}^{N,*}$, both in the $N$-system.*

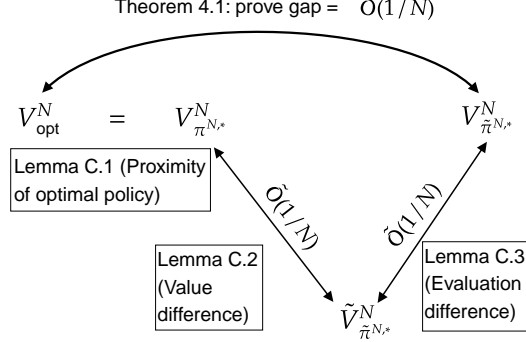

Figure 2: Proof Structure of Theorem 4.1.

A detailed proof of Theorem 4.1 is presented in Appendix C of [33]. Below, we provide an outline and highlight some novel and technically interesting components of the proof. It consists of proving the following statements, as illustrated in Figure 2.

- *Prove that $V_{\mathrm{opt}}^N = V_{\pi^{N,*}}^N$ (Lemma C.1 of [33]), where $\pi^{N,*}$ is the locally optimal policy within the policy class $\Pi_{\delta_N}(\mathbf{y}^*)$.* This is proved by showing that a (globally) optimal policy for the $N$-system belongs to the policy class $\Pi_{\delta_N}(\mathbf{y}^*)$ under Assumption 4.1.

- *Prove that $|V_{\pi^{N,*}}^N(\mathbf{x}, h) - \tilde{V}_{\tilde{\pi}^{N,*}}^N(\mathbf{x}, h)| = \tilde{\mathcal{O}}(1/N)$ (Lemma C.2 of [33]) and $|\tilde{V}_{\tilde{\pi}^{N,*}}^N(\mathbf{x}, h) - V_{\tilde{\pi}^{N,*}}^N(\mathbf{x}, h)| = \tilde{\mathcal{O}}(1/N)$ (Lemma C.3 of [33].* These two lemmas are enabled by the fact that we restrict the policies $\pi^{N,*}$ and $\tilde{\pi}^{N,*}$ to $\Pi_{\delta_N}(\mathbf{y}^*)$, i.e., a $\tilde{\Theta}(1/\sqrt{N})$-neighborhood of $\mathbf{y}^*$, which translates into a Wasserstein distance of $\tilde{\mathcal{O}}(1/N)$ between the respective next-state distributions in the $N$-system and in the Gaussian stochastic system.

We next highlight two interesting components in the proofs.

- *Characterization of optimal policies in the $N$-system.* Lemma C.1 of [33] establishes that under Assumption 4.1, there exists an optimal policy of the $N$-system whose actions are close to the optimal fluid solution $\mathbf{y}^*$. This result is noteworthy because optimal policies of the $N$-system are not well-understood in the literature. Prior work often circumvents this by only studying the LP upper bound $\overline{V}_{\mathrm{LP}}$, which can be loose as shown in Theorem 4.2.

- *Approximate Lipschitz continuity in the Gaussian stochastic system.* A key step in proving Lemmas C.2 and C.3 of [33] is to establish an approximate local Lipschitz property of the value function $\tilde{V}_{\tilde{\pi}^{N,*}}^N$ in the Gaussian stochastic system, where restricting the policy to $\tilde{\pi}^{N,*}$ introduces technical challenges. We overcome these challenges through the careful construction of an action mapping.

## 4.2 The $\Theta(1/\sqrt{N})$ optimality gap of LP-based policies

To complement Theorem 4.1, we next present a result showing that the $\Theta(1/\sqrt{N})$ optimality gap is fundamental to a large class of LP-based policies. Specifically, consider the following policy class, which includes a large class of LP-based policies such as those in [7, 10, 36]:

$$\Pi_{\mathrm{fluid}}(\mathbf{y}^*) := \big\{ \pi \colon \|\pi(\mathbf{x}_h, h) - \mathbf{y}_h^*\|_\infty \leq \kappa \|\mathbf{x}_h - \mathbf{x}_h^*\|_\infty, \ \forall 1 \leq h \leq H \big\}, \tag{21}$$

where $(\mathbf{x}^*, \mathbf{y}^*)$ is an optimal solution of the fluid LP in (8). It has been shown in [10, Theorem 1] that any policy in $\Pi_{\mathrm{fluid}}(\mathbf{y}^*)$ has an $\mathcal{O}(1/\sqrt{N})$ optimality gap. We now show that there exist RMAB instances where this optimality gap order is tight. The result is established based on the example given in Section 3.3, and the detailed proof is presented in Appendix D of [33].

**Theorem 4.2** (Fluid gap). *There exist RMAB instances for which all LP-based policies in $\Pi_{\mathrm{fluid}}(\mathbf{y}^*)$ have an $\Theta(1/\sqrt{N})$ optimality gap; i.e., $V_{\mathrm{opt}}^N(\mathbf{x}_{\mathrm{ini}}, 1) - V_{\mathrm{fluid}}^N(\mathbf{x}_{\mathrm{ini}}, 1) = \Theta(1/\sqrt{N})$, where $V_{\mathrm{opt}}^N$ is the optimal value function, and $V_{\mathrm{fluid}}^N$ is the value function of the optimal policy within the policy class $\Pi_{\mathrm{fluid}}(\mathbf{y}^*)$. Moreover, there is an $\Theta(1/\sqrt{N})$ gap between the optimal value of the $N$-system and the optimal value of the fluid LP in (8); i.e., $\overline{V}_{\mathrm{LP}}(\mathbf{x}_{\mathrm{ini}}, 1) - V_{\mathrm{opt}}^N(\mathbf{x}_{\mathrm{ini}}, 1) = \Theta(1/\sqrt{N})$.*

We remark that although this theorem is proved via a specific example, we believe the result to hold more broadly for many degenerate RMABs. The $\Theta(1/\sqrt{N})$ optimality gap of LP-based policies stems from the $\Theta(1/\sqrt{N})$ approximation error in the fluid approximation, which arises when exchanging the expectation and the minimum operator, as shown in the example in Section 3.3. This phenomenon is common in degenerate RMABs.

## 4.3 Performance improvement

Under the Uniqueness Assumption 4.1, we have shown that our SP-based policy achieves an $\tilde{\mathcal{O}}(1/N)$ optimality gap. When this assumption does not hold, the same optimality gap may not apply. However, Theorem 4.3 below shows that the SP-based policy can still yield improvement over LP-based policies.

Let $\mathbf{y}^*$ be any optimal solution to the fluid LP in (8), and let $\tilde{\pi}^{N,*}$ be the corresponding SP-based policy. Recall that $\tilde{V}_{\tilde{\pi}^{N,*}}^N(\mathbf{x}_{\mathrm{ini}}, 1)$ and $\tilde{Q}_{\tilde{\pi}^{N,*}}^N(\mathbf{x}_{\mathrm{ini}}, \mathbf{y}_1, 1)$ denote the value function and the Q-function of the policy $\tilde{\pi}^{N,*}$ in the Gaussian stochastic system, where in the Q-function action $\mathbf{y}_1$ is applied at time 1. Theorem 4.3 states that if the action given by the SP-based policy $\tilde{\pi}^{N,*}$ at time 1 is "strictly" better than the optimal LP-based action $\mathbf{y}_1^*$ in the Gaussian stochastic system, then the SP-based policy improves over any LP-based policy in $\Pi_{\mathrm{fluid}}(\mathbf{y}^*)$ by $\Omega(1/\sqrt{N})$ in the $N$-system.

**Theorem 4.3** (Performance improvement). *If there exists a positive constant $\epsilon$ independent of $N$ such that $\tilde{V}^N_{\tilde{\pi}^{N,*}}(\mathbf{x}_{\mathrm{ini}}, 1) - \tilde{Q}^N_{\tilde{\pi}^{N,*}}(\mathbf{x}_{\mathrm{ini}}, \mathbf{y}^*_1, 1) \geq \epsilon/\sqrt{N}$, then for any $\pi \in \Pi_{fluid}(\mathbf{y}^*)$, we have $V^N_{\tilde{\pi}^{N,*}}(\mathbf{x}_{\mathrm{ini}}, 1) - V^N_\pi(\mathbf{x}_{\mathrm{ini}}, 1) = \Omega(1/\sqrt{N})$.*

Note that a key strength of this result is that we only need to evaluate the first step action $\mathbf{y}^*_1$ in the Gaussian stochastic system, and the result then holds for all policies in $\Pi_{\mathrm{fluid}}(\mathbf{y}^*)$. We refer to Appendix E of [33] for a detailed proof of Theorem 4.3.

## 5 Numerical experiments

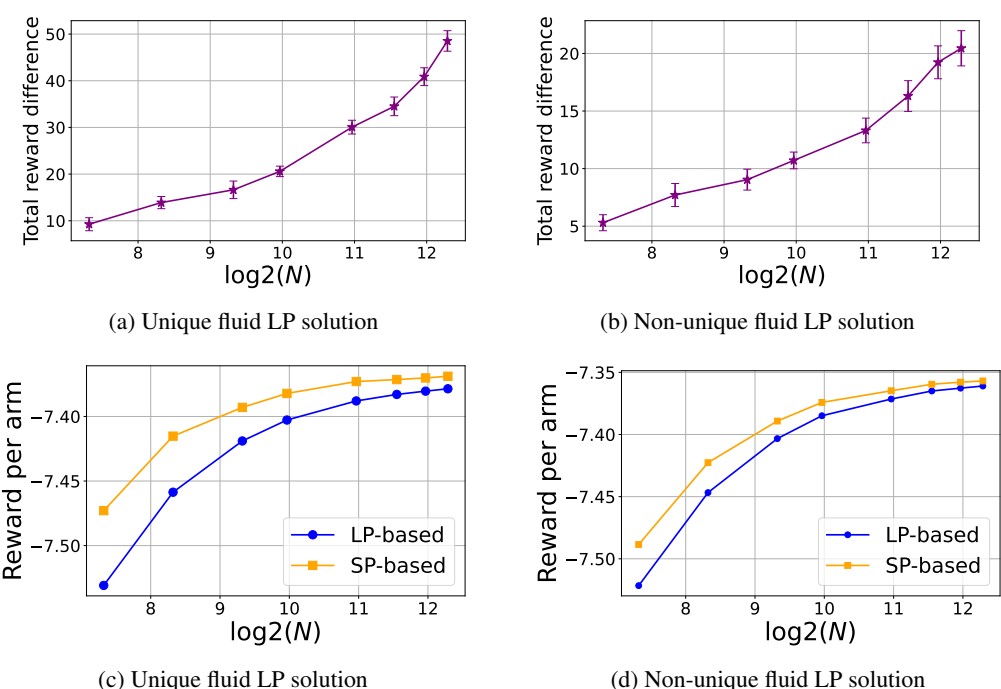

(a) Unique fluid LP solution

(b) Non-unique fluid LP solution

(c) Unique fluid LP solution

(d) Non-unique fluid LP solution

Figure 3: Comparison of LP- and SP-based policies on a machine maintenance example. Top row: total reward difference (SP minus LP) with 2-sigma error bars. Bottom row: reward per arm.

We evaluated the performance of our SP-based policy (Algorithm 1) on a machine maintenance problem [9, 12], an RMAB formulation motivated by real-world trade-offs in preventive maintenance and resource allocation. The resulting SP is solved using the EDDP algorithm [20, Algorithm 3]. The details of the experiments are presented in Appendix B, along with an empirical study of the computational complexity of EDDP for solving the SPs arising from RMABs across varying problem sizes.

We performed two sets of experiments where the problem instances are degenerate: one set where the fluid LP solution is unique (Figure 3a and 3c) and one set where it is not unique (Figure 3b and 3d). For both sets, computing the optimal policies is intractable. In each set of experiments, we evaluated the performance of our SP-based policy and the LP-based policy on a sequence of problems with increasing numbers of machines $N$, and compared the total reward difference. Figure 3 demonstrates that the improvement of our SP-based policy over the LP-based policy grows with $N$ in both settings.

## 6 Conclusion

In this paper, we proposed an SP-based policy for finite-horizon RMABs, leveraging a carefully constructed Gaussian approximation. Motivated by degenerate examples where fluid approximation alone fails to break the $\Theta(1/\sqrt{N})$ gap, we showed that our policy achieves an $\tilde{\mathcal{O}}(1/N)$ optimality gap under the uniqueness assumption, and can achieve $\Omega(\sqrt{N})$ improvement over a large class of LP-based policies.

## Acknowledgments and Disclosure of Funding

The work of Chen Yan and Lei Ying is supported in part by U.S. National Science Foundation (NSF) under grants 2112471, 2134081, 2207548, 2228974, 2240981, 2331780, 2324769; AFOSR grant FA9550-24-1-0002; and Bold Challenges "Accelerate" program from the University of Michigan. The work of Weina Wang is supported in part by U.S. National Science Foundation (NSF) grants ECCS-2145713, CCF-2403194, CCF-2428569, and ECCS-2432545.

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

# A More details of the Gaussian stochastic system and the SP-based policy

**The covariance matrix $\Gamma_h(\mathbf{y}_h^*)$.** Recall that when applying an action vector $\mathbf{y}_h$, the state of the Gaussian stochastic system at the next time step, $\tilde{\mathbf{X}}_{h+1}$, is a random vector of the following form:

$$\tilde{\mathbf{X}}_{h+1} = \mathrm{Proj}_{\Delta^S}\left(\sum_{s,a} y_h(s,a)\mathbf{P}_h(\cdot \mid s,a) + \mathbf{Z}_h/\sqrt{N}\right),$$

where $\mathbf{Z}_h$ is a Gaussian random vector following the distribution $\mathcal{N}(\mathbf{0}, \Gamma_h(\mathbf{y}_h^*))$. The covariance matrix $\Gamma_h(\mathbf{y}_h^*)$ is a constant matrix independent of $N$, and it is defined as follows.

Define, for $(h,s,a) \in [1,H] \times \mathcal{S} \times \mathcal{A}$, the matrix $\mathbf{\Sigma}_h(s,a)$ of size $|\mathcal{S}| \times |\mathcal{S}|$, for which the $(i,j)$-th entry is:

$$\begin{cases} -\mathbf{P}_h(i \mid s,a)\mathbf{P}_h(j \mid s,a), & \text{if } i \neq j \\ \mathbf{P}_h(i \mid s,a)(1 - \mathbf{P}_h(i \mid s,a)), & \text{if } i = j. \end{cases} \tag{22}$$

Then

$$\Gamma_h(\mathbf{y}_h^*) = \sum_{s,a} y_h^*(s,a)\mathbf{\Sigma}_h(s,a). \tag{23}$$

Note that $\Gamma_h(\mathbf{y}_h^*)/N$ is the covariance matrix of the state $\mathbf{X}_{h+1}$ in the $N$-system if the action at time $h$ is $\mathbf{y}_h^*$.

**Rounding procedure.** Throughout this paper, we represent systems and controls as "fractions" of the total population $N$ of arms, using vectors such as $\mathbf{x}, \mathbf{y}$. In the unnormalized-scale system, the coordinates of $N\mathbf{x}$ and $N\mathbf{y}$ therefore need to be integers. If these quantities are not integers, they are implicitly understood to be appropriately rounded via some *rounding procedure*. For example, after obtaining $\mathbf{y}$ from the SP-based policy, we need to applying rounding before using it for the $N$-system.

Formally, a rounding procedure is to solve the following problem: Given an integer $N \in \mathbb{N}$ and $\mathbf{X} \in \Delta_N^S$ so that $N\mathbf{X}$ have integer coordinates, then, for any $\mathbf{Y} \in \mathcal{Y}_{\mathbf{X}}$, we need to construct $\mathbf{Y}^N = \mathrm{round}(\mathbf{Y}) \in \mathcal{Y}_{\mathbf{X}}$ for which $N\mathbf{Y}^N$ have integer coordinates.

This can be accomplished as follows. Set

$$Y^N(s,1) := \frac{\lfloor NY(s,1)\rfloor}{N}, \; Y^N(s,0) := \frac{\lceil NY(s,0)\rceil}{N}, \text{ for } 1 \leq s \leq S-1; \tag{24}$$

$$Y^N(S,1) := \alpha - \sum_{s=1}^{S-1} Y^N(s,1), \; Y^N(S,0) := 1 - \alpha - \sum_{s=1}^{S-1} Y^N(s,0). \tag{25}$$

It is then easy to verify that, $\mathbf{Y}^N := \mathrm{round}(\mathbf{Y})$ defined in (24)-(25) indeed satisfies all these required conditions. Furthermore, by construction, it holds true that

$$\|\mathbf{Y} - \mathrm{round}(\mathbf{Y})\|_\infty \leq \frac{1}{N} = \mathcal{O}(\frac{1}{N}), \tag{26}$$

independently of $\mathbf{X}$ and $\mathbf{Y} \in \mathcal{Y}_{\mathbf{X}}$.

# B More details of numerical experiments

This appendix is a collection of more details on various numerical experiments conducted to confirm the theoretical results of this paper. It is structured as follows:

- Appendix B.1 studies how likely an RMAB instance is degenerate or satisfies the Uniqueness Assumption 4.1.

- Appendix B.2 is centered around numerically solving the SP (14), which appears in Algorithm 1.

- Appendix B.3 displays the parameters and implementation details of the RMAB examples used in Section 5.

All numerical experiments in this paper were conducted on a personal laptop equipped with a 13th Gen Intel(R) i9-13980HX CPU. We note that our experiments are based on solving reasonable size linear programs, for which well-established solution packages are available. We implement the solutions using the Python package **PuLP** and the **Gurobi LP solver**.

## B.1 Degeneracy and Uniqueness Assumption 4.1 in RMAB instances

We conducted a numerical study on how likely a randomly generated RMAB instance is degenerate (Definition 3.1) and the corresponding LP (8) satisfies the Uniqueness Assumption 4.1.

In this experiment, an RMAB instance is generated as follows: Each parameter is sampled via i.i.d. $\exp(1)$ and normalized properly as in the initial condition $\mathbf{x}_{\mathrm{ini}}$ and in the transition kernels $\mathbf{P}_h$. We fix $\alpha = 0.4$ and $H = 5$. We considered two scenarios: "fully-dense" where all entries of a transition kernel are positive; "half-sparse" where half of the entries of each row of a transition kernel are 0. We varied the number of sizes per arm $S = 5, 10, 15, 20$ and tested the degeneracy and the Uniqueness Assumption 4.1 over $10,000$ such RMAB instances, and recorded the numbers in percentage. The results are presented in Table 2.

Table 2: Proportion of RMAB instances satisfying degeneracy and Uniqueness Assumption 4.1

| Transition Kernel | $S$ | Degenerate | Satisfy Uniqueness Assumption 4.1 |
|---|---|---|---|
| Fully-dense | 5 | 11.2% | 100% |
| Fully-dense | 10 | 8.7% | 100% |
| Fully-dense | 15 | 6.1% | 100% |
| Fully-dense | 20 | 5.1% | 100% |
| Half-sparse | 5 | 51.3% | 100% |
| Half-sparse | 10 | 33.3% | 100% |
| Half-sparse | 15 | 28.1% | 100% |
| Half-sparse | 20 | 20.3% | 100% |

We note that overall, degenerate RMABs are a significant proportion among all instances, especially in the half-sparse setting. In addition, all these randomly generated RMABs satisfy the Uniqueness Assumption 4.1.

## B.2 Numerically Solving SP (14)

This section presents the details of the numerical method we used to solve the SP (14). To solve the SP (14), we can transform and simplify it to an $N$-independent and projection-free SP, see Equation (29) of [33] for details. It is a Stochastic Linear Program, which is significantly easier to solve than the original $N$-armed RMAB Problem (4) because the "noise" $\mathbf{Z}_h$'s are predefined Gaussian random vectors whose distributions are independent of the state and action of the system. For such a problem, there exist computationally efficient methods. In our numerical examples, we numerically solved it using the standard Sample Average Approximation (SAA) approach [28, Chapter 5] and the Explorative Dual Dynamic Programming (EDDP) algorithm [20], which is an enhancement of the classical Stochastic Dual Dynamic Programming (SDDP) [26, 27].

We also evaluated the computational complexity of EDDP when varying $H$ and $S$. The results are shown in Figure 4 and include 2-sigma error bars. Each recorded computation time in the figures was averaged from running EDDP on 1,000 RMAB instances, with each instance solved for 10 independent SAA realizations. The results in Figure 4 clearly illustrate that the computational complexity under EDDP increases linearly with respect to the problem parameters $H$ and $S$.

## B.3 Parameters and implementation details of the RMAB examples used in Section 5

The RMAB examples used in Section 5 model a machine maintenance problem. Each machine has five states, where a higher state represents a more deteriorated condition. The first state is a pristine

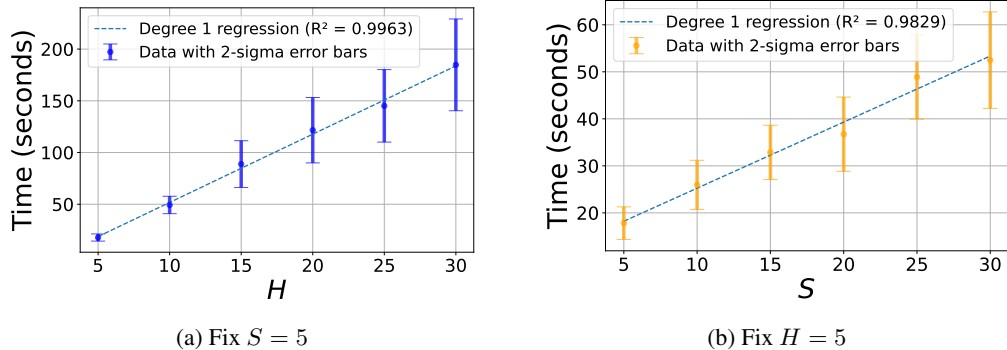

(a) Fix $S = 5$                 (b) Fix $H = 5$

Figure 4: The computation time of EDDP.

state. Under action 1 (performing maintenance), a deteriorated machine has a high probability of returning to the pristine state. Under action 0 (not performing maintenance), the machine gradually deteriorates and has an increasing probability of breaking down as the state worsens. When a breakdown occurs, the machine must be replaced (returning to the pristine state) and incurs a high cost. The negative rewards (i.e., costs) reflect the nature of this model. Note that we can add a sufficiently large common constant to all rewards to offset them, making the values non-negative, as per our reward convention discussed in the main paper.

In these RMAB examples, we set $H = 5$ and $\alpha = 0.4$. We consider arms representing machines each described by a 10 states MDP constructed from two distinct machine types, each having 5 states. The initial state of an arm determines its machine type. Consequently, the transition kernels are structured into two block matrices of size 5. Such a block-structured kernel enables the modeling of multiple machine types under the assumption of homogeneous arms. In other words, the homogeneous RMAB model can be used for heterogeneous systems with a finite number of types. All numerical parameters provided below are recorded with 4 digits of precision.

$$\mathbf{P}(\cdot \mid \cdot, 0) =$$

$$\begin{bmatrix}
0.5415 & 0.4585 & 0 & 0 & 0 & 0 & 0 & 0 & 0 & 0 \\
0.5471 & 0.2265 & 0.2265 & 0 & 0 & 0 & 0 & 0 & 0 & 0 \\
0.7067 & 0 & 0.1467 & 0.1467 & 0 & 0 & 0 & 0 & 0 & 0 \\
0.8578 & 0 & 0 & 0.0711 & 0.0711 & 0 & 0 & 0 & 0 & 0 \\
0.9214 & 0 & 0 & 0 & 0.0786 & 0 & 0 & 0 & 0 & 0 \\
0 & 0 & 0 & 0 & 0 & 0.6396 & 0.3604 & 0 & 0 & 0 \\
0 & 0 & 0 & 0 & 0 & 0.5694 & 0.2153 & 0.2153 & 0 & 0 \\
0 & 0 & 0 & 0 & 0 & 0.6453 & 0 & 0.1773 & 0.1773 & 0 \\
0 & 0 & 0 & 0 & 0 & 0.7007 & 0 & 0 & 0.1496 & 0.1496 \\
0 & 0 & 0 & 0 & 0 & 0.7097 & 0 & 0 & 0 & 0.2903
\end{bmatrix},$$

$$\mathbf{P}(\cdot \mid \cdot, 1) =$$

$$
\begin{bmatrix}
1 & 0 & 0 & 0 & 0 & 0 & 0 & 0 & 0 & 0 \\
0.7337 & 0.2663 & 0 & 0 & 0 & 0 & 0 & 0 & 0 & 0 \\
0.7265 & 0 & 0.2735 & 0 & 0 & 0 & 0 & 0 & 0 & 0 \\
0.6146 & 0 & 0 & 0.3854 & 0 & 0 & 0 & 0 & 0 & 0 \\
0.6054 & 0 & 0 & 0 & 0.3946 & 0 & 0 & 0 & 0 & 0 \\
0 & 0 & 0 & 0 & 0 & 1 & 0 & 0 & 0 & 0 \\
0 & 0 & 0 & 0 & 0 & 0.6037 & 0.3963 & 0 & 0 & 0 \\
0 & 0 & 0 & 0 & 0 & 0.6004 & 0 & 0.3996 & 0 & 0 \\
0 & 0 & 0 & 0 & 0 & 0.7263 & 0 & 0 & 0.2737 & 0 \\
0 & 0 & 0 & 0 & 0 & 0.6138 & 0 & 0 & 0 & 0.3862
\end{bmatrix},
$$

$$
\mathbf{x}_{\text{ini}} = \begin{bmatrix} 0 & 0.5 & 0 & 0 & 0 & 0 & 0.5 & 0 & 0 & 0 \end{bmatrix},
$$

$$
\mathbf{r}(\cdot, 0) = \begin{bmatrix} 0 & -5.4707 & -7.0669 & -8.5784 & -9.2141 & 0 & -5.6942 & -6.4534 & -7.0074 & -7.097 \end{bmatrix}.
$$

In the first set of experiments, we use

$$
\mathbf{r}(\cdot, 1) =
$$
$$
\begin{bmatrix} -1.9963 & -2.085 & -2.035 & -2.0661 & -1.9581 & -1.994 & -1.9647 & -2.2478 & -2.0468 & -2.2821 \end{bmatrix},
$$

which results in a fluid LP with a unique optimal solution. In the second set of experiments, we use

$$
\mathbf{r}(\cdot, 1) = \begin{bmatrix} -2 & -2 & -2 & -2 & -2 & -2 & -2 & -2 & -2 & -2 \end{bmatrix},
$$

which leads to a fluid LP with multiple optimal solutions.

The LP-based policy we compared with is the LP-update policy (also referred to as the LP policy with resolving) proposed in [7, 10]. These studies reported that LP-update is one of the best-performing LP-based policies for finite-horizon RMABs. As mentioned earlier, we used EDDP [20] to obtain the SP-based policy. The predefined policy in Line 12 of Algorithm 1 is specified as follows: If, during the execution of the algorithm, the $N$-system state deviates significantly from the initially selected optimal LP solution — where "significant" is defined as any coordinate exceeding a threshold of 20, a new LP is resolved using the current initial state of the $N$-system and the remaining horizon, similar to the LP-update/resolving policy, and then the first action from the new LP is applied.

In addition, we conducted further tests on several machine maintenance instances, keeping the problem size and structure fixed while varying the reward and transition parameters. All of these instances have unique LP solutions. The observed total reward gaps, shown in Table 3, are of the same order of magnitude as the example in Figure 3a for $N = 100$.

Table 3: Additional experiments for machine maintenace.

| Instance | Total Reward Gap for $N = 100$ Machines (mean $\pm$ std) |
|:---:|:---:|
| 1 | $9.35 \pm 1.21$ |
| 2 | $10.13 \pm 1.53$ |
| 3 | $12.44 \pm 1.97$ |
| 4 | $7.57 \pm 1.35$ |
| 5 | $13.52 \pm 1.56$ |

