# OpenReview forum: "Achieving $\tilde{\mathcal{O}}(1/N)$ Optimality Gap in Restless Bandits through Gaussian Approximation"
_NeurIPS.cc/2025/Conference — NeurIPS 2025 spotlight_

### Official Review · Reviewer_T8Za · 2025-06-29

**Clarity:** 3
**Significance:** 3
**Originality:** 3
**Rating:** 5
**Confidence:** 4

**Summary:**

This paper studies an RMAB problem with $N$ homogeneous arms. Prior results based on LP relaxation may result in a larger $\Theta(1/\sqrt{N})$ optimality gap when the RMAB problem is degenerate. In contrast, this paper proposes a new Gaussian approximation and stochastic-programming (SP) based policy that can achieve a smaller $\Theta(1/N)$ optimality gap, which only requires the relaxed solution to be unique. A small 2-state problem with 2 time-steps is used to show the performance advantage of the SP-based policy.

**Questions:**

1. My main concern is how the SP-based policy can be solved efficiently. The example in the paper involves only 2 per-agent states and 2 time-steps, which is quite small and less practical. With only two per-agent states, the occupancy measure at each time-step can be parameterized by 1 value. Then, backward induction over 2 time-steps can be written as an (expectation) optimization problem over just one variable (in Line 207). This optimization problem still needs to be solved numerically, but is not that difficult. For a general problem with $K$ states, the occupancy measure will form a $K-1$-dimensional hyperplane. With multiple steps, the backward induction will lead to a highly-complex optimization problem in $K-1$ dimension. The paper does not seem to provide a way to solve this general backward induction under Gaussian approximation.

Further, note that one of the main advantages of LP relaxation (and other related approaches such as Whittle index) is the ability to decompose the problem, which is crucial for overcoming the curse of dimensionality. Specifically, for an $N$-arm problem, solving the whole RMAB involves a backward induction over $O(N^K)$ global states (if we again use the occupancy measure to represent a global state). In contrast, solving the decomposed problem only involves backward induction over $K$ per-agent states, which is much easier. It is unclear how the Gaussian approximation and SP-based policy can benefit from similar decomposition. If such a decomposition is not possible, then even approximating the general backward induction described above will incur complexity exponential in $K$. If that is the case, the Gaussian approximation will have little advantage over the standard backward induction for the original problem in terms of complexity reduction.

**Ethical Concerns:**

["NO or VERY MINOR ethics concerns only"]

**Final Justification:**

The rebuttal partially addresses my earlier concerns on the solution complexity of the Gaussian approximation. Even though the complexity is still significantly higher than the fluid limit (with decomposition), at least it seems more manageable. Thus, I increase my rating to Accept. It would be useful to include this complexity discussion in the main body of the paper.

**Limitations:**

yes

**Paper Formatting Concerns:**

None.

**Quality:**

3

**Strengths And Weaknesses:**

Strength:

1. The idea of using Gaussian approximation to refine the LP-relaxation seems interesting.

2. The result on the $O(1/N)$ optimality gap is a strong.

Weakness:

1. It is unclear how to solve the SP-based policy efficiently.

---

> ### Author Rebuttal · Authors · 2025-07-29
>
> We greatly appreciate the positive comments from the reviewer. We next address the computation issue of SP versus the backward induction suggested by the reviewer.
>
> First, we would like to clarify that our approach applies to a general number of states, time horizon, and number of arms.
> The stochastic program (SP) (15)-(17) is for this general setting, and a variety of existing algorithms can be used to solve this SP.
> In our numerical experiments (Section 5), we used the EDDP algorithm Lan (2022) to solve the SP, and our experiment was for a setting with $K=10$ states per arm and a time horizon of $H=5$. In Figure 5 (Appendix G.3 in the supplementary material), we evaluated the time complexity of EDDP with RMAB size as large as $K=5, H=30$ and $K=30, H=5$.
>
> Next, we would like to discuss in detail the computational complexity.
> Indeed, directly solving the backward induction incurs a complexity of at least $\Theta(N^K)$, which is exponential in the number of states $K$. In contrast, the SP we formulated allows for more efficient algorithms.
> At an intuitive level, although the SP doesn't exactly decompose the problem in the same way as the LP does, it still takes advantage of state aggregation.
> More concretely, our Gaussian approximation aggregates the randomness in state transitions into a Gaussian noise ($\mathbf{Z}_h$).
> One of our key ideas is to make the noises ($\mathbf{Z}_h$'s) independent of actions.
> This allows the SP to be solved through the Sample Average Approximation (SAA) technique, which uses random samples of the $\mathbf{Z}_h$'s and reduces the problem into a deterministic linear program.
> The number of random samples required is polynomial in $N$ and $K$.
> One straightforward approach then is to solve this SAA-induced linear program using the interior point method (IPM) (see, e.g., recent results on the complexity of IPM in Cohen and Song (2021)), which can already lead to a computation complexity that is polynomial in both $N$ and $K$.
>
> That said, directly solving the resulting SAA-induced linear program can lead to an exponential dependence on the time horizon.
> In our paper, we chose to use the recently developed EDDP algorithm to solve the SAA.
> In our experiments, EDDP solves the SP efficiently, and in these numerical examples, the computational complexity is linear in both the time horizon and the number of states $K$ (Figure 5 in Appendix G.2 in the supplementary material).
>
>
> **References**
>
> - Cohen, M. B., Lee, Y. T., and Song, Z. (2021). "Solving linear programs in the current matrix multiplication time." *Journal of the ACM (JACM)*, 68(1), 1–39.
>
> - Lan, G. (2022). "Complexity of stochastic dual dynamic programming." *Mathematical Programming*, 191(2), 717–754.

---

> > ### Comment · Reviewer_T8Za · 2025-08-03
> >
> > I wish to thank the authors for the clarification on the solution complexity! The response addresses my earlier concerns. I will elevate my rating to accept.

---

> ### Author Response · Authors · 2025-08-05
> **Thanks!**
>
> We are delighted to know that we addressed your question. Thanks again for your great questions, and thanks for revising the rating!

---

### Official Review · Reviewer_2XWR · 2025-07-02

**Clarity:** 3
**Significance:** 3
**Originality:** 3
**Rating:** 5
**Confidence:** 4

**Summary:**

In this paper, the authors have studied the finite horizon Restless Multi-armed Bandit (RMAB) problem under the homogeneous arm setting. Existing works on RMAB show that Linear-Programming (LP) based policies derived from the fluid approximation can result in an exponentially small optimality gap, however, under the assumption that the RMAB satisfies a non-degeneracy condition. However, in reality, RMABs are often degenerate, and in that case, LP-based policies can result in $\Theta(1/\sqrt{N})$ optimality gap per arm. The authors in this paper have proposed a new Stochastic Programming (SP) based policy that can achieve an $O(1/N)$ optimality gap for degenerate RMABs (for the first time) under the uniqueness assumption. The authors claim that the construction of a Gaussian stochastic system can capture the mean and variance of the RMAB dynamics, and it is more accurate than the fluid approximation.

**Questions:**

1.	What is the motivation behind considering a Gaussian stochastic system as an approximation of the original system? What other kinds of distribution can be chosen to offer a good approximation? Does a Gaussian stochastic system offer the best approximation?
2.	Why is there a need of a $1/\sqrt{N}$ factor multiplying $Z_h$ in equation (12)?
3.	The meaning of the policy class described in (13) is not clear. What is the physical interpretation behind this class construction?
 I wonder whether it would include a large number of feasible policies. The main result in the paper (Theorem 4.1) heavily depends on this class of policies. However, the significance of the result can be better understood if we get more insight about the class and its cardinality.
4.      Please explain what is meant by the round operator in Algorithm 1.
5.	Theorem 4.1 holds with rounding. Is it possible to extend this theorem to the case without rounding too?
6.	Theorem 4.3 assumes the existence of a positive constant $\epsilon$ independent of $N$. How can we guarantee the existence of such an $\epsilon$?

**Ethical Concerns:**

["NO or VERY MINOR ethics concerns only"]

**Final Justification:**

Authors have addressed the concerns raised by me (such as regarding construction of policy class and rounding) and provided satisfactory response. It will be good if the authors could include the additional simulation results in the main paper.

**Limitations:**

Yes. However, in the conclusion section, limitations of the proposed approach can be explicitly mentioned for more clarity.

**Quality:**

3

**Strengths And Weaknesses:**

The paper is well-written and technically solid. However, more intuitions need to be provided to understand the impact of the result better. More insight is needed on the construction of policy class, as the main results of the paper are based on this class construction. The simulation section can be enriched with simulations on more scenarios.

---

> ### Author Rebuttal · Authors · 2025-07-29
>
> We hope our responses to the reviewer's questions below provide clearer intuition and deeper insight into the policy class construction and related issues. We also appreciate the suggestion to enrich the simulation section with additional scenarios.
> To address this suggestion, we have conducted additional tests on several machine maintenance instances, keeping the problem size and structure the same while varying model parameters for rewards and transition probabilities. All these instances exhibit uniqueness in their LP solutions. The observed total reward gaps in these experiments, displayed in the table below, remain within the same order of magnitude as the example presented in the left subfigure of Figure 3 (Section 5 of the paper) for $N = 100$. We plan to incorporate these additional results, along with the corresponding model parameters, in a future revision of our paper.
>
> **Table: Total reward gap across additional machine maintenance instances**
>
> | Instance | Total Reward Gap for $N=100$ Machines (mean ± std) |
> |----------|-----------------------------------------------------|
> | 1        | 9.35 ± 1.21                                          |
> | 2        | 10.13 ± 1.53                                         |
> | 3        | 12.44 ± 1.97                                         |
> | 4        | 7.57 ± 1.35                                          |
> | 5        | 13.52 ± 1.56                                         |
>
>
> - **What is the motivation behind considering a Gaussian stochastic system as an approximation of the original system? What other kinds of distribution can be chosen to offer a good approximation? Does a Gaussian stochastic system offer the best approximation?**
>
> We consider the Gaussian distribution because by the central limit theorem, the Gaussian distribution is the limiting distribution of the normalized mean of $N$ independent random variables.  The central limit theorem also implies that other kinds of distributions can't be such a limiting distribution, so the Gaussian approximation offers the best approximation asymptotically. In particular, the Wasserstein distance bound (Lemma D.1) may not hold under other distributions.
>
> - **Why is there a need of a $1/\sqrt{N}$ factor multiplying $\mathbf{Z}_h$ in equation (12)?**
>
> The factor $1/\sqrt{N}$ again arises from the central limit theorem (CLT).
>      To see this more clearly, consider $N$ independent and identically distributed random variables $X_1,X_2,\dots,X_N$, each with mean $\mu$ and variance $\sigma^2$. By the CLT, as $N$ becomes large, we have the following approximation
>      \begin{equation*}
>          \frac{\sum_{i=1}^N X_i-N\mu}{\sigma\sqrt{N}}\approx Z,
>      \end{equation*}
>      where $Z$ is a standard Gaussian random variable.
>      Rewriting this gives
>      \begin{equation*}
>        \frac{1}{N}\sum_{i=1}^{N}X_i \approx \mu + \frac{\sigma}{\sqrt{N}}Z,
>      \end{equation*}
>      where there is a $1/\sqrt{N}$ factor multiplying $Z$.
>      The $1/\sqrt{N}$ factor in our equation (12) comes from the same reason, although for a more complicated setting.
>
>
> - **The meaning of the policy class described in (13) is not clear. What is the physical interpretation behind this class construction? I wonder whether it would include a large number of feasible policies. The main result in the paper (Theorem 4.1) heavily depends on this class of policies. However, the significance of the result can be better understood if we get more insight about the class and its cardinality.**
>
> Roughly speaking, the policies in (13) are policies such that when the system state $\mathbf{x}_h$ is in an $\tilde{\mathcal{O}}(1/\sqrt{N})$ neighborhood of the optimal fluid state $\mathbf{x}^{\star}_h$, the action taken is also within an $\tilde{\mathcal{O}}(1/\sqrt{N})$ neighborhood of the optimal fluid action $\mathbf{y}^{\star}_h$. We would like to emphasize that this policy class includes commonly studied fluid-based policies in the literature, e.g., the policies in references Zhang and Frazier (2021), Gast et al. (2023), and Brown and Zhang (2023); and most existing policies for RMAB are fluid-based.  Moreover, it also includes the optimal policy for the $N$-armed problem when the uniqueness assumption (Assumption 4.1) is satisfied.
>
> - **Please explain what is meant by the round operator in Algorithm 1.**
>
>
> The rounding operation guarantees that the action generated by the algorithm pulls an *integer* number of arms. It is needed due to the following reason: when computing the first- and second-order approximations, we extend the state and action spaces from their original discrete sets (with granularity $1/N$) to continuous-valued probability simplices. The action computed in this continuous domain must subsequently be converted back to the discrete domain, ensuring that all values multiplied by $N$ are integers and thus applicable to the $N$-armed problem. This rounding procedure is explicitly detailed in Appendix C of the supplementary material.
>
>
> - **Theorem 4.1 holds with rounding. Is it possible to extend this theorem to the case without rounding too?**
>
> Without rounding, the policy may output an action that pulls a fraction of an arm, which is typically not allowed in a restless bandit problem. That said, if pulling a fraction of an arm is allowed (say 0.5 arm), e.g. by using a probabilistic method (say pull one arm with probability 0.5), then Theorem 4.1 holds without rounding.
>
>
> - **Theorem 4.3 assumes the existence of a positive constant $\varepsilon$ independent of $N$. How can we guarantee the existence of such an $\varepsilon$?**
>
>
> We expect that such an $\varepsilon$ exists when the fluid optimal action $\mathbf{y}_1^{\star}$ at time step $1$ is not an optimal action for the Gaussian stochastic system.
> To see this, consider the $N$-independent stochastic program (107), which is derived from problem (14) through re-centering and scaling, as detailed in Appendix G.2 of the supplementary material. If the optimal first-stage solution of (107) differs from zero, which translates to $\mathbf{y}_1^{\star}$ not being an optimal solution, then $\varepsilon$ corresponds to the gap between the optimal objective value of (107) and the objective value (or the $Q$-value) obtained by evaluating the fluid-optimal first-stage action. Equivalently, the value of $\varepsilon$ is the difference between the largest Q-value and the Q-value under the fluid action. If the fluid action is sub-optimal, then the gap $\varepsilon$ is strictly positive and independent of $N$. Otherwise, the fluid optimal action $\mathbf{y}_1^{\star}$ is optimal for the Gaussian stochastic system, and our second-order policy will take the fluid optimal action.
>
>
> **References**
>
> - Brown, D. B., and Zhang, J. (2023). "Fluid policies, reoptimization, and performance guarantees in dynamic resource allocation." *Operations Research*.
>
> - Gast, N., Gaujal, B., and Yan, C. (2023). "Linear program-based policies for restless bandits: Necessary and sufficient conditions for (exponentially fast) asymptotic optimality." *Mathematics of Operations Research*.
>
> - Zhang, X., and Frazier, P. I. (2021). "Restless bandits with many arms: Beating the central limit theorem." arXiv preprint arXiv:2107.11911.

---

> > ### Comment · Reviewer_2XWR · 2025-08-04
> >
> > Thanks to the authors for their detailed response which has answered my queries. I am happy to raise my score.

---

> > > ### Author Response · Authors · 2025-08-05
> > > **Thanks!**
> > >
> > > We are delighted to know that our rebuttal adequately addressed your questions. Thanks again for your great questions, and thanks for revising the rating!

---

### Official Review · Reviewer_yu2p · 2025-07-03

**Clarity:** 3
**Significance:** 4
**Originality:** 4
**Rating:** 5
**Confidence:** 4

**Summary:**

Authors study the restless multi-armed bandit problem in the finite horizon setting. They give a new policy and improved theoretical optimality gap in the degenerate setting, based on Gaussian approximation applied to the standard fluid approximation. They support their findings with empirical results.

**Questions:**

Since the policy is based on the fluid approximation, it would be helpful to give some idea of how much the SG policy can deviate from the base fluid approximation and still retain optimality or feasibility of computation time – what is the factor at play there?

Can the gaussian approximation technique be extended to infinite horizon settings?

Can you comment on how you might relax the homogeneity assumption and still maintain the gap improvement?

Comment on computational complexity of the approach – how it scales with |S| and H?

Can this approach be extended to the multi-action/WCMDP setting?

**Ethical Concerns:**

["NO or VERY MINOR ethics concerns only"]

**Final Justification:**

Key result in RMAB literature of interest to a wide audience. Strong results and writing in the original submission, as well as responses to all reviewer concerns and clarifications.

**Quality:**

4

**Strengths And Weaknesses:**

Strengths:
* Key new result in RMAB literature, improving the optimality gap while relaxing the non-degeneracy assumption
* Achieved through a new technique based on Gaussian approximation, building on established methods based on fluid approximation
* Detailed illustrative example effectively demonstrates the mathematical intuition

Weaknesses:
* assumes homogenous arms (i.e., all arms have the same MDP probas and rewards)
* lacking comments on the scalability of the approach with respect to the size of the state space and the length of the horizon
* additional motivation for the degenerate setting would improve the significance -- are there convincing real-world examples?

---

> ### Author Rebuttal · Authors · 2025-07-29
>
> We greatly appreciate the reviewer's positive comments. We next address the weaknesses and questions raised by the reviewer.
>
> - **Assumes homogeneous arms (i.e., identical transition probabilities and rewards across arms).**
>
> Our results can be directly applied to settings with limited heterogeneity. A more detailed response can be found in the answer to the reviewer's question on extending to heterogeneous cases.
>
> - **Lacking comments on scalability of the approach with respect to the state-space size and horizon length.**
>
> We provide a detailed discussion on scalability later when answering the reviewer's question on scalability,  highlighting computational complexity and empirical runtime performance.
>
> - **Additional motivation for the degenerate setting would improve significance—are there convincing real-world examples?**
>
> Another real-world example is the applicant screening problem studied in Brown and Smith (2020), which explicitly exhibits degeneracy by requiring non-trivial tie-breaking. We will add this example to the revision.
>
>
> - **Since the policy is based on the fluid approximation, it would be helpful to give some idea of how much the SG policy can deviate from the base fluid approximation and still retain optimality or feasibility of computation time – what is the factor at play there?**
>
> We thank the reviewer for this insightful question. Our second-order Gaussian policy specifically seeks improvements by considering corrections within a neighborhood of size $\tilde{\mathcal{O}}(1/\sqrt{N})$ around the fluid-optimal state and action. When the LP has a unique solution, we proved in the paper that the optimal policy of the $N$-arm problem is within $\tilde{\mathcal{O}}(1/\sqrt{N})$ neighborhood of the LP (fluid) solution (Lemma D.1 in the supplementary material). Therefore, limiting the deviation within this   $\tilde{\mathcal{O}}(1/\sqrt{N})$ neighborhood retains the optimal policy and also guarantees that our second-order approximation has $\tilde{\mathcal{O}}(1/N)$ approximation error (in terms of Wasserstein distance). Expanding the neighborhood would increase the second-order approximation error; while shrinking the neighborhood would exclude the optimal policy. So it turns out that $\tilde{\mathcal{O}}(1/\sqrt{N})$ is the ``exact'' deviation needed for our proposed method. We will add this discussion to the revision.
>
> - **Can the gaussian approximation technique be extended to infinite horizon settings?**
>
> The algorithm can be directly extended to the infinite horizon settings by considering a second-order approximation around the LP solution to the infinite horizon problem. However, the $\tilde{\mathcal{O}}(1/N)$ optimality gap would require a different proof due to the fundamental difference between finite-horizon MDPs and infinite-horizon average-reward MDPs.
>
> - **Can you comment on how you might relax the homogeneity assumption and still maintain the gap improvement?**
>
> The heterogeneous setting is more challenging because the arms can no longer be aggregated into a single representative arm as in the homogeneous case. However, if the arms can be divided into different types, and the number of types is independent of $N$, then our result can directly apply. In fact,  our machine maintenance problem in Section 5 is such a heterogeneous setting and considers two types of machines (see details in Appendix G.3 of the supplementary materials). We will clarify this in the revision. The ``fully'' heterogeneous setting would likely require new techniques.
>
> - **Comment on computational complexity of the approach – how it scales with $|S|$ and $H$?**
>
> The computational complexity of our approach stems primarily on solving the Gaussian SP in Line 6 of Algorithm 1. The numerical solution methods are detailed in Appendix G.2 in the supplemented material: After recentering and scaling, we obtain an $N$-independent multi-stage linear stochastic program (problem (107) in the supplementary material). The computational complexity of solving problem (107) depends on the number of states per arm, $|S|$, and the horizon length, $H$. Our numerical evaluations using Explorative Dual Dynamic Programming (EDDP, Lan (2022)) indicate that the runtime scales linearly in both $|S|$ and $H$, as illustrated in Figure 5 in the supplementary material.
>
> - **Can this approach be extended to the multi-action/WCMDP setting?**
>
> The key is to come up with an equivalent version of the action mapping algorithm in Section D.2.3 (supplementary materials) and then prove it is locally Lipschitz. The proof in this paper utilizes some unique properties of the RMAB. If a similar result can be proven for the more general WCMDP setting, then the result should generalize.
>
>
> $\textbf{References:}$
>
> - Brown, D. B., and Smith, J. E. (2020). "Index policies and performance bounds for dynamic selection problems." Management Science, 66, 3029–3050.
>
> - Lan, G. (2022). "Complexity of stochastic dual dynamic programming." Mathematical Programming, 191(2), 717–754.

---

> > ### Comment · Reviewer_yu2p · 2025-08-05
> > **Response to authors**
> >
> > I've read all the reviews and authors' responses. I appreciate their thoughtful answers to my questions about the mathematical intuition of the key result as well as thoughts on several ways to attack relaxing key assumptions. I also appreciate the insightful question and author discussion on the diffusion approximation and agree that adding the extended discussion to the paper will enrich what is already a strong submission.

---

> > > ### Author Response · Authors · 2025-08-06
> > > **Thanks!**
> > >
> > > We are delighted that we answered your questions. Thanks again for your great questions and positive comments!

---

### Official Review · Reviewer_5dNH · 2025-07-03

**Clarity:** 3
**Significance:** 3
**Originality:** 3
**Rating:** 5
**Confidence:** 3

**Summary:**

This paper studies a finite-horizon N-armed restless bandit problem with homogenous arms (identical state space and transition kernel across arms). The authors propose a Gaussian approximation-based approach that tracks the mean as well as variance of the RMAB process, resulting in a more accurate representation of the limiting process compared to standard fluid limit-based approaches. The resulting stochastic programming-based policy is shown to improve sub-optimality gap per arm by a factor of $1/\sqrt{N}$ over LP-based policies (resulting from fluid approximation) in the case of degenerate RMABs.

**Questions:**

None, please see the previous section.

**Ethical Concerns:**

["NO or VERY MINOR ethics concerns only"]

**Final Justification:**

The contributions became clearer after the authors disambiguated their (simpler) Gaussian approximation from classical diffusion scaling. There are novel elements in their approach and analysis that I believe can benefit the wider community.

**Limitations:**

Yes

**Quality:**

3

**Strengths And Weaknesses:**

Proofs were not carefully checked but the intuition provided is sound and generally easy to follow. The general strengths of this paper in my opinion are:

1. Well-written and instructive to follow.
2. That a higher order approximation leads to a significant improvement (breaking the $1/\sqrt{N}$ barrier) is nice to see.

Not a weakness, but a question/suggestion: The proposed Gaussian approximation bears resemblance to "Diffusion Approximation" techniques common in Operations Research and Stats literature. It would be good if the authors can comment on this and make connections to similar techniques that might have appeared under different names. I am happy to revise my score upwards pending resolution of this point. In particular, I want to be sure that this (or a very similar) problem isn't already addressed in adjacent literature under different technical nomenclature.

Also curious to hear the authors' thoughts on whether the $\mathcal{O}\left( 1/N \right)$ gap can be further reduced (under the same assumptions).

---

> ### Author Rebuttal · Authors · 2025-07-29
>
> We greatly appreciate the reviewer's willingness to reevaluate the score based on our response. We next address the connection to diffusion approximation and whether the $\mathcal{O}(1/N)$ gap can be further reduced.
>
> **Connection to diffusion approximation:**
>
> In the current version of the paper, we only briefly mention the connection to diffusion approximation in Appendix B of the supplementary material. We agree that this connection warrants further elaboration, and we plan to expand this discussion following the outline below.
>
> First, there is a rich literature on diffusion approximations for continuous-time queueing systems, particularly in the heavy-traffic regime (see, e.g., seminal work [3], [4], [12], [13] in the reference list of the supplementary material).
> This body of work typically analyzes queueing systems under a *fixed policy*, and the focus is on characterizing the convergence or the rate of convergence of the centered and scaled queueing system to the diffusion limit.
> In contrast, our work addresses a different goal for a different application: we focus on finding a near-optimal policy for a discrete-time restless bandit problem.
>
> A more closely related line of work, which builds on the diffusion approximation literature, is on approximate diffusion control for queueing systems or stochastic control in general (see, e.g., seminal work and textbook such as Beneš et al. (1980), Harrison (1985), and Harrison and Wein (1989)). In this line of work, the problem is approximated by a diffusion control problem, which is then solved via the associated Hamilton-Jacobi-Bellman (HJB) equation.
> Recent work by Braverman et al (2020) shows that the HJB equation naturally arises under the second-order approximation of value functions of general Markov decision processes (MDPs).
>
> While we can apply the same diffusion control to our problem, solving the HJB equation is known to be notoriously difficult in general. This is because the diffusion coefficients often depend on the control, which makes the HJB a highly nonlinear PDE. This actually motivated us to consider a different second-order approximation.
>
> A key innovation in our work is to replace (state, action)-dependent diffusion terms with (state, action)-independent diffusion terms. We construct our Gaussian approximation system such that the covariances of the Gaussian noises (analogous to the diffusion coefficients) are independent of the actions, by leveraging the optimal fluid solution.
> This proposed Gaussian approximation allows us to tap into stochastic programming techniques, where (state,action)-independent randomness is required for efficient algorithms like the EDDP algorithm we used in this paper. In other words, our Gaussian approximation can be viewed as a further ``approximation/simplification'' of the diffusion approximation, but it has the same order of approximation error in our problem and can be solved efficiently using existing stochastic programming methods.
>
>
> We note that our results are possible because in the restless bandit problem, the noise terms are on a smaller scale compared to the deterministic terms as the number of arms scales, due to the central limit theorem.
> This is reflected in the $1/\sqrt{N}$ factor for $\mathbf{Z}_h$ in the SP in equations (14)-(17).
> This scale separation makes it possible to use the (state,action)-independent covariances generated by the fluid optimal solution as a surrogate for the action-dependent covariances, and leads to a critical result (Lemma D.1) that shows that the optimal policy of the $N$-arm problem is within $\tilde{\mathcal{O}}(1/\sqrt{N})$ neighborhood of the fluid solution under a uniqueness assumption. We do not usually have such a property in traditional diffusion control.
>
>
> In summary, our approach may be viewed as a further approximation of the traditional ``diffusion approximation''. Based on the proximity result (Lemma D.1) for restless bandits, our approximation reduces the problem to simpler stochastic programming (instead of solving nonlinear PDEs) and guarantees  $\tilde{\mathcal{O}}(1/N)$ optimality gap.
>
> We hope these expanded discussion addresses the reviewer's concern, and we will incorporate a detailed version of this discussion into our paper.
>
> **The $\mathcal{O}(1/N)$ gap.**
>
> We thank the reviewer for this question. Our conjecture is that any asymptotic methods that reduce the problem to continuous state and action spaces will have an $\Omega(1/N)$ gap in a general setting. This is because rounding procedure (like the one detailed in Appendix C), which is necessary to map a continuous action to a discrete action in the N-arm system, will result in an $\mathcal{O}(1/N)$ error in general, except for some special cases, such as when the optimal policy is a priority policy.
>
>
> **References:**
>
> - Beneš, V. E., Shepp, L. A., and Witsenhausen, H. S. (1980). "Some solvable stochastic control problems." Stochastics, 4(1), 39–83.
>
> - Braverman, A., Gurvich, I., and Huang, J. (2020). "On the Taylor expansion of value functions." Operations Research, 68(2), 631–654.
>
> - Harrison, J. M. (1985). "Brownian Motion and Stochastic Flow Systems." Wiley, New York.
>
> - Harrison, J. M., and Wein, L. M. (1989). "Scheduling networks of queues: Heavy traffic analysis of a simple open network." Queueing Systems, 5(4), 265–279.

---

> > ### Comment · Reviewer_5dNH · 2025-08-06
> >
> > I thank the authors for the detailed explanation. I would really like to see some of this discussion moved to the main paper itself. In particular, I believe it is a very interesting finding that a simpler second order approximation with state/action-$independent$ coefficients suffices to achieve O(1/N) sub-optimality gap, where classical diffusion scaling is analytically intractable. Further, if the authors can formally show that the O(1/N) rate is essentially unimprovable (and it is okay to prove it in a contrived setting if warranted), I think that would only go on to strengthen the paper. I'm revising my rating and vote for acceptance.

---

> > > ### Author Response · Authors · 2025-08-06
> > > **Thanks!**
> > >
> > > We thank the reviewer again for the great questions/suggestions. Also, thanks for revising the rating. We will incorporate the discussions into the main paper as the reviewer suggested.

---

### Decision · Program_Chairs · 2025-09-17

**Decision:**

Accept (spotlight)

**Comment:**

The paper tackles the degenerate restless MAB in the finite horizon setting with $N$ homogenous arms. While it is widely accepted in prior work that the optimality gap in this setting that can be achieved with LP-based policies is $\Theta(1/\sqrt{N})$, the authors surprisingly proves that using a stochastic-programming-based policy, this optimality gap can be reduced to $\tilde{O}(1/N)$.

The reviewers agree the paper is instructional and well written and the results are novel, surprising and significant as they improve on widely accepted limit. The authors obtain this key new result by way of a novel Gaussian approximation method (see reviewer yu2p's comments). While this Gaussian approximation bears resemblance to diffusion approximation, the authors detail the contrast of their method versus diffusion approximation in their reply to reviewer 5dNH's comments.

The reviewers suggest the submission would be even stronger if it incorporated the essence of the comparison to diffusion approximation, a discussion on the scalability of the approach with respect to the size of the state space and the length of the horizon and provided convincing real-world examples of the utility of the setting.

Nonetheless, the paper represents a strong technical contribution with the potential to spur wide interest in the community, and it is my and the reviewer's opinion that this paper should be accepted.